# Chemical Characterization and Seasonality of Ambient Particles (PM_2.5_) in the City Centre of Addis Ababa

**DOI:** 10.3390/ijerph17196998

**Published:** 2020-09-24

**Authors:** Worku Tefera, Abera Kumie, Kiros Berhane, Frank Gilliland, Alexandra Lai, Piyaporn Sricharoenvech, Jonathan Samet, Jonathan Patz, James J. Schauer

**Affiliations:** 1School of Public Health, College of Health Sciences, Addis Ababa University, Addis Ababa 9086, Ethiopia; aberakumie2@yahoo.com or; 2Department of Biostatistics, Columbia University, New York, NY 10032, USA; ktb2132@cumc.columbia.edu; 3Keck School of Medicine, University of Southern California, Los Angeles, CA 90033, USA; gillilan@usc.edu; 4Environmental Chemistry and Technology Program, University of Wisconsin-Madison, Madison, WI 53706, USA; alexandra.lai@weizmann.ac.il (A.L.); sricharoenve@wisc.edu (P.S.); jjschauer@wisc.edu (J.J.S.); 5Office of the Dean, Colorado School of Public Health, Aurora, CO 80045, USA; jon.samet@CUAnschutz.edu; 6Global Health Institute, University of Wisconsin, Madison, WI 53706, USA; patz@wisc.edu; 7Wisconsin State Hygiene Laboratory, University of Wisconsin-Madison, Madison, WI 53706, USA

**Keywords:** fine particulate matter, organic matter, organic carbon, elemental carbon, dust

## Abstract

Ambient air pollution is a growing public health concern in major African cities, including Addis Ababa (Ethiopia), where little information is available on fine particulate matter (PM_2.5_, with aerodynamic diameter <2.5 µm) pollution. This paper aims to characterize annual PM_2.5_, including bulk composition and seasonal patterns, in Addis Ababa. We collected 24-h PM_2.5_ samples in the central city every 6 days from November 2015 to November 2016. The mean (±SD) daily PM_2.5_ concentration was 53.8 (±25.0) µg/m^3^, with 90% of sampled days exceeding the World Health Organization’s guidelines. Principal components were organic matter (OM, 44.5%), elemental carbon (EC, 25.4%), soil dust (13.5%), and SNA (sulfate, nitrate, and ammonium ions, 8.2%). Higher PM_2.5_ concentrations were observed during the heavy rain season, while crustal dust concentrations ranged from 2.9 to 37.6%, with higher levels during dry months. Meteorological variables, vehicle emissions, biomass fuels, unpaved roads, and construction activity contribute to poor air quality. Compared to the Air Quality Index (AQI), 31% and 36% of observed days were unhealthy for everyone and unhealthy for sensitive groups, respectively. We recommend adopting effective prevention strategies and pursuing research on vehicle emissions, biomass burning, and dust control to curb air pollution in the city.

## 1. Introduction

The Global Burden of Disease estimates for 2017 show that more than 4.9 million people die prematurely every year due to the adverse health impacts of air pollution from all causes, while ambient particulate matter pollution exposure alone accounts for 4.6 million deaths [1]. Airborne particulate matter, size-fractionated as PM_10_ and PM_2.5_, with aerodynamic diameter less than 10 µm and 2.5 µm, respectively, is a widely used indicator of ambient air pollution [2]. Particles in the PM_2.5_ size range reach the deepest portions of the respiratory system, including the small airways and alveoli [3,4,5].

The effects of PM_10_ and PM_2.5_ on human health have been extensively studied with regard to increased morbidity and excess mortality [6,7]. The adverse health effects are both acute and chronic and reflect various underlying mechanisms, including inflammation and carcinogenesis. Particles are diverse in their characteristics, including chemical composition. The chemical composition of PM_2.5_ has been linked to the biological impacts of PM_2.5_ inhalation, such as oxidative potential [8,9]. Therefore, investigating the components of PM_2.5_ in polluted urban air is necessary to understand the links between the health risks posed by PM_2.5_ and the potential toxic components. Characterization of PM_2.5_ is also needed for source apportionment, which leads to the design of more efficient control strategies and air pollution abatement policy actions [10].

Sub-Saharan Africa (SSA), in general, and the Eastern Africa region in particular, where Ethiopia is located, lack urban air quality monitors for PM_2.5_ and other pollutants. Hence, there is a paucity of evidence on fine particulate air pollution [11,12]. Addis Ababa is one of the largest metropolitan cities in SSA, characterized by a rapidly growing population with a high rate of urbanization. The number of motorized vehicles per capita is still fairly low, at 3 vehicles per 1000 persons [13]; however, the majority of the vehicles are quite old, with most operating in the absence of an emission standard, modern emission control technology, or an effective smog controlling enforcement system. Moreover, narrow roads coupled with lax annual vehicle safety inspections result in a substantial number of high-emitting trucks, buses, and automobiles. In addition, the open burning of solid waste, combustion of diesel for power generation, and particulate matter emissions from industries and construction activities also contribute to poor air quality in the city. Given these many air pollution sources, this study examines the levels and sources of ambient air pollution, particularly PM_2.5_, to help design appropriate policy interventions and control mechanisms in order to reduce the impact on population health and the environment.

## 2. Materials and Methods

### 2.1. Study Setting

Addis Ababa, a city with a population of about 4.7 million [14], is located at an average altitude of 2355 m above sea level. Geographically, Addis Ababa is surrounded by mountain chains extending from the northeast to the western part of the city, with plains in the east and south. Consequently, the winds drive emissions towards the western and northwestern areas of the city. Temperature inversions are common during the seasons with low temperatures, especially during early morning and evening [12].

The weather in Addis Ababa is categorized into three main seasons based on precipitation: the Dry season (from October to January, and also May); the light to moderate rain season—Wet 1 (from February–April); and the main rain season—Wet 2 (from June–September) [15]. According to the National Meteorological Agency, the temperature in Addis Ababa ranges from a monthly average minimum of 8 degrees C to a maximum of 25 degrees C. The prevailing wind direction from the annual Wind-rose diagram shows that the sampling location has an easterly wind direction, whereas a station located near the Bole area has a prevailing southeasterly wind direction. The monthly mean rainfall in Addis Ababa ranges from a minimum of 7 mm (Nov./Dec.) to 280/290 mm (July/August). The Relative Humidity (%) monthly mean ranges from a minimum of 45.5% in December to a maximum of 79.5% in July. The monthly average daylight ranges from 11.6 to 12.6 h; monthly average sunshine ranges from 3 h (July/August–Wet 2 season) to 9 h (December–Dry season) [16]. Appendix A) depict line plots of daily minimum temperature, maximum temperature, and precipitation during 2015–2016 at the study site. 

### 2.2. Sampling Approach

Integrated 24-h PM_2.5_ samples were collected at a central city meteorological station (“Met Station”) (9.019046° N, 38.747360° E), once every 6 days from November 2015 to November 2016. The sampling station was located near the meteorological measuring instruments above the ground at a height of 2 m. The Met Station is the oldest meteorological station in Ethiopia, also serving as the site for ambient air quality monitoring in Addis Ababa city; it continuously records measurements of pollutant gases (NO_x_, CO, and O_3_). The HOBO^®^ data logger (Model Part # H08-032-08; Onset Computer Co., Bourne, MA, USA) was installed near the sampling hood under the shed of the PM_2.5_ monitor to measure relative humidity (RH) and temperature over the 24-h sampling period. The station is located in a typical downtown urban area surrounded by residential houses as well as public institutions and private businesses. The area is situated near one of the city’s major traffic networks, and includes an adjacent busy road (at about one-half km distance from the site) leading to the city’s largest open market area (‘*Merkato’*), located over a steep hill. There is an adjoining road (minimum of 25 m from the monitor). According to the near-road monitoring guidance of the US Environmental Protection Agency (US EPA), a monitoring station should be at a minimum distance of 10–20 m from traffic [17] Appendix A depicts the map of monitoring stations in Addis Ababa, Ethiopia. Samples were collected on quartz fiber (37 mm diameter, GE Whatman, USA., and Teflon filters (37 mm diameter, PTFE Teflo^TM^ Pall Corporation, Port Washington, New York USA, concurrently using two 5-stage Harvard Cascade Impactor samplers (Harvard University Chan School of Public Health, Boston, MA 02115, USA), which were provided by the Southern California Environmental Health Sciences Center (SCEHSC), University of Southern California (USC, Los Angeles, CA, USA) [18]. The samplers were operated at a flow rate of 5 LPM. The flow rates were measured before and after each sample using an Omega Rotameter (Omega Engineering Inc., Norwalk, CT 06854, USA), which was calibrated using a Gilibrator ‘Bubble meter’ (Gilibrator-2 Calibrator, MPN 800271 (120 V), Sensidyne Industrial Hygiene and Safety Instrumentation, St. Petersburg, FL 33716, USA). The mean flow rate over the sampling period was used to calculate the sampled air volume. Quartz filters were baked for at least 12 h at 550 °C before sampling and stored in prebaked foil both before and after sampling. Field blanks were also collected for every 10 set of samples. To prevent evaporation of volatile components, all collected samples were sealed in polystyrene Petri dishes and then stored frozen, below −20.0 °C before analysis.

### 2.3. Laboratory Analysis

The PM samples were shipped for mass and chemical components analysis to the Water Science and Engineering Laboratory and Wisconsin State Laboratory for Hygiene at the University of Wisconsin-Madison. The collected particle mass concentrations were determined by weighing the Teflon filters before and after sampling on a microbalance (MT 5, Mettler-Toledo Inc., Highstown, NJ, USA) in a controlled room with the temperature kept at 20.0 °C ± 2.0 °C, and relative humidity at 40.0% ± 2.0%. A static neutralizer (500uCi Po210, NRD LLC, Grand Island, NY, USA) was used to eliminate electrostatic charges. To equilibrate the filters with humidity levels, the filters stayed in the weighing room for at least 12 h before weighing. An ion chromatography (IC, Dionex ICS 2100 for anions and Dionex 1100 for cations; Thermo Fisher Scientific, Waltham, MA, USA) method was used for analyzing inorganic ions after one-quarter of each Teflon filter was extracted in high-purity water. Inorganic ions, including chloride, nitrate, sulfate, sodium, ammonium, calcium, and potassium, were analyzed by ion chromatography (ICS 1100 and 2100, Dionex, Imperial, PA 15126, USA) [19]. The extracts were also used to measure the water-soluble organic carbon (WSOC) content of the samples using a Total Organic Carbon (TOC) analyzer (Siever M9, GE Analytical Instruments, Boulder, CO, USA). To determine the elemental and organic carbon levels [20,21], a 1.0 cm^2^ punch from the quartz fiber filters was analyzed using thermal-optical transmittance technique (Sunset laboratory, Forest Grove, OR, USA). The subtraction of the WSOC concentration from the organic carbon (OC) concentrations in the sample yielded the water-insoluble OC (WIOC) [22,23]. Total trace element concentrations were analyzed using one-quarter of each Teflon filter, and quantified after digestion and subsequent analysis by high resolution sector field inductively coupled plasma-mass spectrometry (SF-ICP-MS) [24]. Three elements—gallium, indium, and bismuth—were used as internal standards for the HR-ICP-MS analysis along with authentic standards for each quantified element.

### 2.4. Data QA/QC

The samples were collected using quartz and Teflon filters and Harvard Cascade Impactor samplers at a flow rate of 5.0 LPM based on a standard protocol (adopted from University of Southern California (USC) Children’s Health Study). Field and laboratory blanks, check standards, spikes, and duplicate samples were run to ensure and control the quality of data for each chemical analysis. Spike and standard recoveries were 88.7% on average for all ions measured by IC (ranging from 69.4% to 99.7% for individual ions) and 78.4% on average for WSOC. ECOC sucrose spike recoveries were 90.9% to 105.0%, depending on the analysis batch (using standard reference materials by the US National Institute of Standards and Technology with certified reference values for given elements). A total of 18 field blank filters were used for QA/QC. On average among all analyses, field blank concentrations were less than 10% of sample concentrations. Duplicate precision agreement (relative standard deviation of duplicate measurements) was better than 5% on average for all analyses. Average field blank values were used to blank-correct all reported sample concentrations, and uncertainties were calculated based on standard deviation of field blanks and estimated instrument precision. Filter data were considered as valid if the flow rate did not vary by more than 5% of 5 LPM.

### 2.5. Chemical Mass Closure

We used chemical mass closure to assess major mass constituents of the collected particles in Addis Ababa. The calculated chemical constituents are ammonium, nitrate, sulfate, dust oxides, elemental carbon (EC), particulate organic matter (POM), and other elements [25,26]. The soil dust oxide concentrations were determined by converting trace element data to the most common form of oxides presented in the crustal soil composition (i.e., Al_2_O_3_, SiO_2_, CaO, K_2_O, FeO, Fe_2_O3, MnO_2_, MgO, and TiO_2_) using corresponding molecular weight conversion factors [27]. POM was estimated from the measured OC values multiplied by a conversion factor of 1.4 [28,29,30]. The concentrations of the other elements were calculated from the sum of all the trace elements (except major soil), which mostly represent the anthropogenic and heavy metal content of aerosols.

### 2.6. Data Analysis

The final PM_2.5_ mass concentration was obtained by subtracting the average concentration of field blanks from the measured mass concentration. The particulate OM was derived as a product of OC by a factor of 1.4. For descriptive analysis, we used mean, standard deviations, minimum and maximum. Graphical presentation of the data employed Microsoft Excel (2011, Microsoft Corporation, Redmond, WA 98052, USA) and R-project tools (R version 3.5.2, https://www.r-project.org/about.html) to depict graphs using line graphs, bar charts, and box plots.

## 3. Results

### 3.1. PM_2.5_ Concentration

Sixty-one sampling sets (days) of valid data for PM_2.5_ concentration and composition were available from a total of 69 sets collected and then evaluated through the QA/QC procedure. The annual average (±SD) PM_2.5_ mass concentration in central Addis Ababa during November 2015 to November 2016 was 53.8 (±25.0) µg/m^3^ and the mean (±SD) of monthly average concentrations during the same period was 55.1 (±17.7). The PM_2.5_ daily concentration observed in this study ranges from 19.1 to 127.0 µg/m^3^.

### 3.2. PM_2.5_ Bulk Composition

Table 1 and Figure 1 summarize the monthly and mean annual levels of PM_2.5_ and the elemental mass concentrations, respectively. On average, the dominant components of fine particles by proportion were OM (43.0%), EC (25.5%), dust (13.0%), sulfate (5.7%), ammonium (1.9%), nitrate (0.6%), and other ions (1.0%), including Na^+^, Cl^−^, ws K^+^, and Ca^2+^. These components accounted for an average of 88.0% of the PM_2.5_ mass.

The particulate OM, formed from compounds of hydrogen, oxygen, nitrogen, and sulfur in addition to OC, was the dominant component of PM_2.5_ with contributions of 26.0–63.0% for most days of the year. The monthly average PM_2.5_ concentration shows that OM ranged from 32.9% to 57.3% (Appendix A). The carbonaceous components, including EC, OC, WSOC, and WIOC, and dust contributions, which were found abundantly in PM_2.5_ are addressed in the following sections.

Other abundant forms of PM_2.5_ components were the secondary inorganic ions of sulfate, nitrate, and ammonium, which are formed in the atmosphere through a natural process of photochemical oxidation or condensation of the precursor gases of SO_2_, NH_3_, and NO_x_ [31]. The annual average sulfate contribution was 5.7%. Whereas the daily SO_4_ components were less than 3.0% of PM_2.5_ mass in the majority of samples (70%), higher proportions, from 4–10%, were observed mostly during September, October, and November 2016, with monthly averages of about 6%, 7%, and 4%, respectively.

Nitrates in the urban atmosphere are formed from NOx emissions from motor vehicle exhausts and are often in the form of ammoniated nitrate (NH_4_-NO_3_). The PM_2.5_ in Addis Ababa had an overall daily mean NO_3_ concentration of 0.7% (with daily contribution to PM_2.5_ varying from 0 to 5% and a monthly average of 0.1–2.3%). Ammonium ions also comprised 1.7% of the annual mean PM_2.5_ levels. Total secondary ion concentrations were slightly higher during the dry season than at other times of the year. Other ions, such as Cl^−^, Na^+^, Ca^2+^, and K^+^ made small contributions in total, from very low to ~3% in monthly mean values.

### 3.3. Organic and Elemental Carbon

The carbonaceous component made up a major fraction of the PM_2.5_ in Addis Ababa. Organic matter (OM) was the most abundant species of PM_2.5_. Together, OM and EC comprised 69.8 ± 9.3% of the daily levels of PM_2.5_, ranging from 46 to 94%. The daily mean (SD) concentrations of OM (OC*1.4) and EC were 23.2 ± 11.0 µg/m^3^ and 13.7 ± 6.8 µg/m^3^, respectively. The lowest monthly mean of OM proportion was observed during October 2016, at 25.5%, followed by May and August 2016, accounting for ~27% of PM_2.5_ mass (Appendix A).

Generally, the proportion of OM in PM_2.5_ varied seasonally, with the lowest occurring from August through November 2016. Conversely, in contrast to OM, the proportion of EC reached the maximum level (27–30%) during the heavy rain season (Wet 2), while comprising a consistently lower proportion during the other seasons (18–22%).

The findings from this study revealed that the mean ratio of OC/EC is 1.3 (±0.5) and it ranges from 0.7 to 2.7. Water-soluble OC (WSOC) contributes with mean WSOC/OC of 0.22 (±0.18), ranging from negligible to 0.84.

Water soluble and insoluble components of OC (WSOC/WIOC) and EC levels, along with the seasonal trend, are presented in Figure 2a. The majority of OC was formed from WIOC, comprising 78% of the daily mean of OC (i.e., daily mean concentration of WIOC, 12.9 ± 6.7 µg/m^3^). The monthly mean (SD) level of WSOC was 3.6 ± 3.1 µg/m^3^, ranging from undetectable levels to 10.9 µg/m^3^ in August 2016, which had the highest daily WSOC observation, after one extreme measurement in May and another two measurements in September 2016 were excluded. A spike in the concentrations of water soluble OC (WSOC) was observed during three observation days (10 May, 19 September and 25 2016) compared to concentrations of the respective OC levels, and the spike coincided with a religious ceremony which uses the burning of biomass, particularly tree branches and leaves, as part of the celebration. As it appears that these levels overestimated the concentration of WSOC, they were excluded from the analysis. Figure 2b shows moderate correlation between OC and EC, suggesting primary source particle pollutants dominate.

There was relatively large seasonal variation in PM_2.5_ mass concentration, which was significantly higher during the heavy rain (Wet 2) season (June–September (78.6 ± 8.2 µg/m^3^) than during the light to moderate (Wet 1) season (February–April), and May (dry) (46.3 ± 5.4 µg/m^3^), as well as the main dry season (Dry) (October to January), which had the lowest observation.

Daily mean PM_2.5_ mass concentrations were available from two monitors maintained by the US Department of State using US EPA equivalent reference methods. The “Central” monitor was located in the northern part of the city (background area) while the “School” monitor was located in an inner-city site in southwestern Addis Ababa. PM_2.5_ was measured with Beta Attenuation Monitors (BAMs), beginning in August 2016 at the two sites and present study site (Figure 3a). Our results were compared with these BAM data, as shown in Figure 3a. The figure shows a similar seasonal pattern of PM_2.5_ concentrations at all three sites with higher levels during the wet seasons compared to the remainder of the year.

Figure 3b, from an aggregated data of the two BAM monitors, shows the diurnal pattern of PM_2.5_, with two peaks: during the morning rush (8:00–9:00 GMT+3) and late afternoon (extended from 19:00–22:00 GMT+3).

### 3.4. Dust Contributions and Components

Oxides in dust (Al_2_O_3_, Fe_2_O_3_, TiO_2_, MgO, CaO, K_2_O, MnO_2_, and SiO_2_) contributed a daily average of 13.5% of PM_2.5_, with the highest contribution in February (37.6%), followed by two more peak days during March (30.6%) and May (32.8%), which are in the light rain (Wet 1) and dry seasons (a transitional period between the two wet seasons), respectively (Figure 4a). The monthly mean contributions of dust were 25.0%, 24.0% and 22.0% during February, March, and May, respectively. The lowest proportion of dust was observed during the heavy rain season (Wet 2), with mean daily (2.9%) and monthly (5.0%) dust contributions to PM were observed during August. The daily variation of the dust oxides over the study period is depicted in Figure 4a,b, as a percentage contribution to—and dust oxides of—PM_2.5_, respectively.

### 3.5. Trace Elements and Heavy Metals

As shown in Figure 5, Cd, As, Pb, Mn, and Zn are among the toxic metals of public health importance analyzed in this study. Pb, Mn, and Zn have notable concentrations, with Zn having the highest mean levels. The mean (SD) Pb level was 8.4 ± 5.3 ng/m^3^; ranging from 1.7 to 68.0 ng/m^3^. Most of the toxic elements of health significance in crustal materials, including Se, Sb, Ag, S, Sn, Pb, As, and Cu, found in PM_2.5_ were found to be significantly enriched, beyond what is naturally occurring in geological materials [32], by anthropogenic sources in Addis Ababa. Appendix A, listed the crustal enrichment factor (CEF) of trace elements—with CEF>100 shown in bold.

## 4. Discussion

In this study, 61 24-h PM_2.5_ filter samples were collected successfully over one year at a central location in Addis Ababa, Ethiopia. We estimated that the annual PM_2.5_ concentration was about 5-fold higher than the WHO annual mean Air Quality Guideline (AQG) value of 10.0 µg/m^3^ [33]. The annual mean PM_2.5_ air quality standard of the Ethiopian EPA (15.0 µg/m^3^) was also exceeded by over 3-fold [34]. The highest particulate matter concentrations were during the heavy rain season (Wet 2), suggesting the importance of increased fuel burning for heating during this period. Compared to the daily mean AQG value of the WHO (25.0 µg/m^3^) and that of the US EPA standard for PM_2.5_ (35.0 µg/m^3^), the PM_2.5_ levels in Addis Ababa exceeded these values by over 90.0% and 76.0% on the sampling days, respectively. The levels of PM_2.5_ measured in the city center of Addis Ababa were higher than at the two more peripheral sites maintained by the US Department of State. However, day-to-day variation was quite similar across the three sites.

In Ethiopia, there have been few studies to date on PM_10_; we did not identify any studies on ambient PM_2.5_ [11,12,35]. The health effects of air pollution have not yet been well studied in Ethiopia. However, a recent literature review by Tarekegn and Gulilat [36] discussed air pollution and health using secondary data sources in Addis Ababa. The disease burden trend in the city from 2003 to 2017 showed that acute upper respiratory infection and chronic obstructive pulmonary disease both increased by an annual rate of one-half and pneumonia grew by nearly a quarter every year. This increase could largely be attributed to the increasing traffic-related air pollution.

Moreover, despite the limited sample size, a calculated air quality index (AQI) from the observed data based on the US EPA classification (AQI: Good = 0–50; Moderate = 51–100; Unhealthy for Sensitive Groups—USG = 101–150; Unhealthy = 151–200; Very Unhealthy = 201–300; Hazardous = 301–500) shows that 36% and 31% of the observed days fell under USG and Unhealthy categories, respectively, while only a third of the observed days were Moderate.

Previous studies elsewhere show that WSOC is highly correlated with reactive oxygen species (ROS) activity [9,22]. Hence, the dominance of carbonaceous matter in PM_2.5_ composition, which is likely emitted from motor vehicles and biomass burning sources, could be linked to the respiratory health of the population in the city [36].

The WHO global estimate based on satellite-driven data indicated that the median level of PM_2.5_ in urban areas of Ethiopia was 36.0 µg/m^3^, ranging from 10.0 to 132.0 µg/m^3^. The range is in good agreement with this study. However, the annual average in urban Ethiopia is, obviously, lower compared to the PM_2.5_ mass concentration in this study, 53.8 µg/m^3^, which is largely comprised of organic and elemental carbonaceous materials indicating the importance of primary emissions from vehicles and burning of biomass fuel. The total number of vehicles in Ethiopia for 2015 was 587,400, based on data from Deloitte Africa Automotive Insights Report 2016 [37]. While the overall quantity of vehicles in the country is very small (3 per 1000 people) compared to other countries, particulate matter pollution in the metropolitan city of Addis Ababa was found to be high, possibly reflecting a contribution from motor vehicles that are mostly likely old and poorly maintained.

Brown et al. discussed the variability of the OM/OC ratio from the measurement of concentrations and concluded that using constant ratios for determining seasonal or daily concentration may over or underestimate the concentration of OM—hence, the health impact of exposure from PM_2.5_, of which OM is a major component [38]. For this study, we used OM/OC ratio of 1.4 [30].

Snyder et al. [39] argued that estimates of secondary OC using EC-tracer method could be potentially flawed due to presence of non-biomass sources of WSOC. In this study, we did not estimate the secondary organic aerosols. Further analysis on carbonaceous contents of PM_2.5_ is needed to accurately determine the temporal variability between primary and secondary OC [23].

Additionally, the contribution of biomass burning is also substantial. A large proportion of homes in urban areas nationally use biomass fuel (70.5%), including charcoal (30.0%) [40]. Although only 3.3% of the households in Ethiopia’s urban areas have a car or truck [41], the majority of these motor vehicles are found in or near the city of Addis Ababa [41].

Compared to studies in the northern and sub-Saharan regions of Africa (SSA), this study shows a mass concentration PM_2.5_ at least two times higher than those in studies conducted from 2008 to 2010 in Nairobi (Kenya), at both urban and industrial background sites [42], as well as in a semirural site in Accra (Ghana) [43]. Nonetheless, similar findings were reported from a study conducted in the urban center of Cairo (Egypt) during 2012 [44].

A seasonal 2-weeks sampling campaign conducted during the summer of 2011 in 7 locations of Jeddah, Saudi Arabia, reported an overall average of PM_2.5_ mass concentration of 28.4 ± 25.4 μg/m^3^, with the highest mass concentration recorded in a suburban site (73.2 ± 65.1 μg/m^3^) and the highest PM_2.5_/PM_10_ ratio (0.52) compared to other sites in the city; the residential and urban sites had considerably lower mass concentration during the same season. The suburban site PM_2.5_ mass concentration in Jeddah (Saudi Arabia) [45] was comparable to the our results in Addis Ababa, with similar seasonal average mass concentrations. This shows that, despite variation in pollutant sources, PM_2.5_ mass concentration in this study is as high as those metropolitan cities in developing countries with large vehicle numbers and population size.

For comparison, Sharma and Mandal reported that the average mass concentration of fine PM was 125.5 ± 77.2 μg/m^3^ (range: 31.1–429.5 μg/m^3^) in Delhi (India), which is one of the most polluted cities in Asia. In Delhi, particulate organic matter (OM) is the highest contributor to PM_2.5_ mass, with 27.5%, followed by soli/crustal matter (16.1%), ammonium sulfate, and ammonium nitrate contributing 16.1% and 13.1%, respectively, while sea salt and light-absorbing carbon account for 17.1% and 10.2%, respectively [46]. The present study reported a higher OM and soil dust contribution to PM_2.5_ but a lower proportion of ammoniated compounds (ammonium nitrate and ammonium sulfate) compared to the Delhi study [46].

A similar finding was also reported in Beijing (China) during 2010 [47]; though it is one of the Asian cities, traditionally known for having relatively high pollution levels in the region. The daily levels of PM_2.5_ in this study are also higher than a recent study in Tehran (Iran) at 33.0 ± 11.0 µg/m^3^. However, the proportion of non-attainment days in this study, compared to the WHO guideline values, is similar to that in Tehran, with over 91.0% of the days exceeding WHO guidelines. Higher levels of PM in African cities, however, have been reported. Studies on the city center in Ouagadougou, Burkina Faso [48], and on two low income neighborhoods in Nairobi (Kenya) in 2013 — Korogocho, 166.0 µg/m^3^ and Viwandani, 67.0 µg/m^3^ [42]—show that the annual mean PM_2.5_ concentration was much higher than the levels observed in this study.

Summer (June–Sept) in central Ethiopia is a season characterized by high rainfall, which might be expected to reduce the PM_2.5_ levels, especially in July and August. We suggest that the higher levels during this season could result from motor vehicle exhaust and increased biomass burning activity for cooking as well as space heating, accentuated by frequent temperature inversions during days when mornings are cold. Yet the level of crustal dust is higher during the dry season of the year compared with the wet season: in fact, the lowest level of crustal dust was observed in August. This may be due to rainfall that washes the atmosphere of suspended dust and the clearing of dust from the unpaved roads and asphalt streets. Moreover, construction activities, especially excavation of ground soil, are significantly reduced during this time of the year. The regional contribution of pollutants, particularly desert dust driven into the city of Addis Ababa by its prevailing eastward winds, needs to be further investigated in atmospheric modeling studies.

An earlier study in Addis Ababa [35] reported that 34.0% to 66.0% of PM_10_ mass concentrations were sourced from geological materials, while EC and OC in urban locations contributed, 31.0% and 60.0% of PM_10_ mass, respectively. In this study, we found an equivalent proportion of EC and OC contributions to PM_10_ mass compared to the prior study [35], while both Ethiopian studies seemed to have much higher levels of OC and EC compared to studies in the US and Europe, as well as in far east Asian cities such as Tehran [49,50,51,52]. This is largely due to the very low contribution of secondary inorganic aerosol, including sulfate, nitrate, and ammonium (SNA), and sea salts, compared to other regions of the world. The results emphasize the importance of direct primary emissions of particulate matter from carbon-containing fuels and that secondary inorganic components are less of a contributor to PM_2.5_ in Addis Ababa.

A review of PM_2.5_ composition by Zhang et al. [53] showed that for many of the megacities in Asia, the US, and Europe, sulfate, nitrate, and ammonium contribute more than 50%. This includes cities with high pollution levels such as Beijing; there are some exceptions, however, such as Mexico City, where the contribution represents about a third. Yet the contribution of SNA to PM_2.5_ is higher during winter compared to summer [53]. The present study, however, found that SNA contributes less than 10% to PM_2.5_ composition. In a globally and spatially diversified study [54], ammoniated sulfate in Kanpur (India), was the highest contributor in the range and more than double the sulfate observed in our study. While the combination of ammoniated sulfate and ammonium nitrate generally contribute about a quarter of PM_2.5_ composition, SNA only added up to 8.1% in the current study.

The two dominant sources of water-soluble organic carbon (WSOC) are secondary organic aerosols (SOA) and the burning of biomass fuels [25]. The peak WSOC levels that occurred during September and May 2016 might be due to events related to public holidays that take place across the city, involving massive burning of biomass fuel, including eucalyptus woods. Nonetheless, the WSOC seems higher during the wet seasons than the dry season.

Secondary sulfate is a major component of urban fine particulate matter in countries where coal or other high-sulfur fuels are used [49,55]. The sources of particulate matter sulfate in this study are likely high sulfur diesel vehicle fuel and diesel power generators used as alternative power sources during frequent grid power outages.

The World Health Organization reported that lead (Pb) exposure from environmental sources, including the air we breathe, accounted for an estimated 0.6% of the global burden of disease, with the highest burden occurring in the developing world [56]. The total concentration of lead (Pb) found in this study is similar to the results of a study in the city center of Ouagadougou (Burkina Faso) [47]. Compared to the levels in the urban center of Cairo (Egypt) [44], where leaded gasoline was sold on the market, our study shows a 10-fold lesser concentration. This might be attributable to the ban on leaded gasoline in Ethiopia since 2004 and to other industrial sources of lead in Cairo. However, the concentration of lead (Pb) in a semirural area of Accra (Ghana) [43], reported from measurements during 2006 and 2007, indicated much lower levels by a factor of 3 than those in this study. None of the airborne PM_2.5_ lead concentrations observed in this study exceeded the World Health Organization guideline values of 0.5 µg/m^3^ (annual average) or approached 0.1 µg/m^3^. Etyemezian et al. [12] reported that lead levels never exceeded 0.1 µg/m^3^ in PM_10_, which was observed 6 months after the phase-out of leaded gasoline in Addis Ababa. However, Teju et al. [57] reported a very high lead concentration from roadside soil samples compared to a control site soil samples in Addis Ababa, which might be due to past deposition of leaded gasoline and other anthropogenic sources.

## 5. Conclusions

This study is one of the few studies to use ground monitor data to determine the levels and chemical species composition of PM_2.5_ in Addis Ababa. The findings of this study will shed light on an ongoing effort to improve air quality in the region by providing critical evidence on ambient urban air pollution, particularly on levels, bulk composition, and seasonal variations of the components of PM_2.5_. The results from this study also serve as input to apportion resources for ongoing research to identify the problems and to take measured action based on the evidence. The PM_2.5_ mass concentration levels in the urban center of Addis Ababa exceeded both the WHO guidelines (10 µg/m^3^) and the Ethiopian EPA air quality standards (15 µg/m^3^) for the annual mean. These findings have a public health implication, as two-thirds of the days observed in this study were either unhealthy for the general population or unhealthy for sensitive groups (children, asthmatics, pregnant women and people with pre-existing conditions) based on the US EPA air quality index (AQI). The measurements taken at the urban center site in Addis Ababa show that PM_2.5_ mass concentration varies by seasonality and meteorological events, and are highest during the heavy rain season (Wet 2). Organic matter (OM), elemental carbon (EC), and dust make up more than three-quarters of the annual average PM_2.5_ concentration, with carbonaceous matters contributing the largest portion, followed by soil dust, with the wet season having a clear edge in increased concentration. Primary organic carbon (OC) dominates the carbonaceous contribution during the wet season. Therefore, we can conclude that anthropogenic sources, including the use of unclean fuel for vehicles, biofuels for cooking food at home, and uncontrolled dust in the urban environment could have human health consequences and are causes of concern for public health.

To guide policy for reducing ambient particulate air pollution in Addis Ababa, we recommend adopting effective prevention strategies and promoting research on vehicle emissions, biomass burning, including waste, and dust control to curb air pollution in the city.

## Figures and Tables

**Figure 1 ijerph-17-06998-f001:**
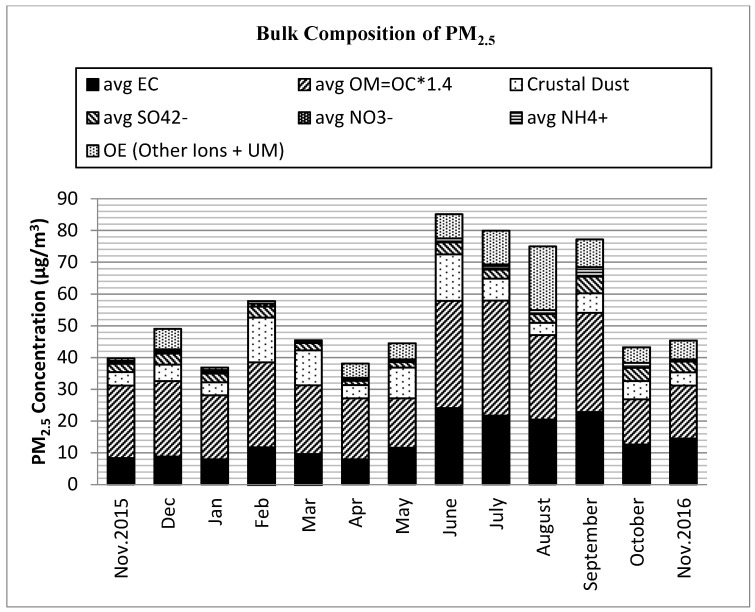
Particulate matter (PM_2.5_) chemical species by concentration (µg/m^3^) of major components in Addis Ababa; EC = Elemental carbon. OM = Organic matter. OE = Other elements. Other Ions = ws Na^+^, ws Cl^−^, ws K^+^, ws Ca^2+^. Other = Unidentified matter.

**Figure 2 ijerph-17-06998-f002:**
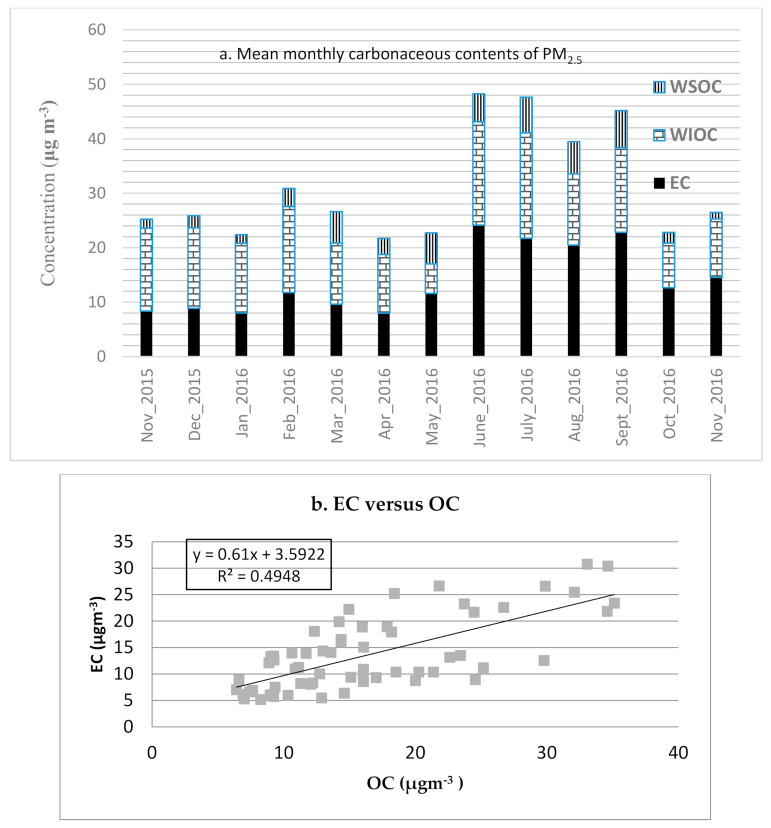
(**a**) Carbonaceous contents of PM_2.5_ in central Addis Ababa, October 2015–December 2016. (**b**) Scatter plot of OC versus EC concentrations of PM_2.5_ in central Addis Ababa, October 2015–December 2016. N.B.: 24-December 2015; 16-May; 9-June, 15-June; and 3-July 2016 were missing data due to various reasons (including power outages), not days with zero values. Extreme values during 10-May, 19-September, and 25-September 2016 were excluded (not shown on Figure 2a as data aggregation to monthly).

**Figure 3 ijerph-17-06998-f003:**
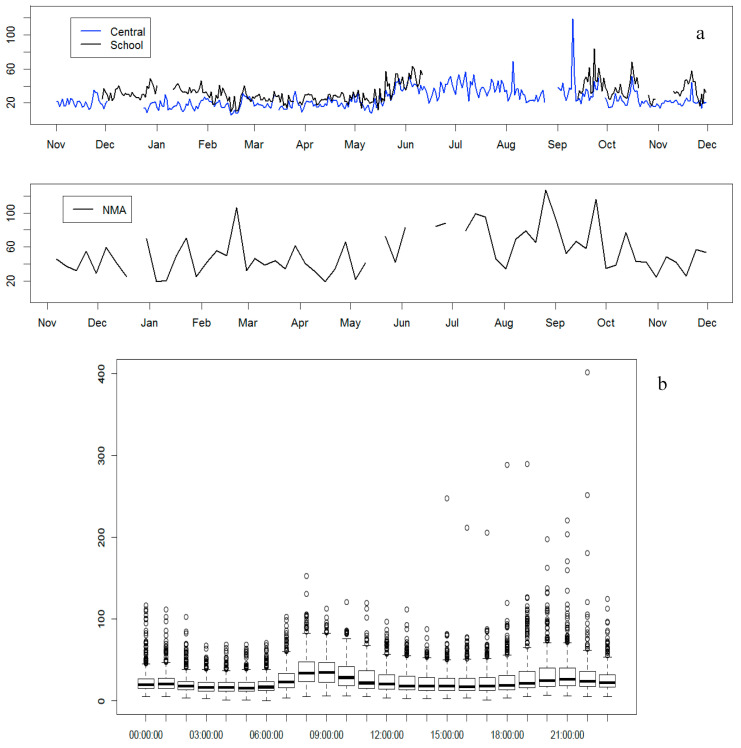
(**a**) Line graph of PM2.5 concentration (µg/m^3^) trend during all seasonality (i) during August 2016 through December 2017 (US Embassy sites: Blue line—Central and Black line—School) and (ii) during November. 2015–November 2016 (NMA—Met Office) site) in Addis Ababa. (**b**) Diurnal pattern of the concentration of PM_2.5_ (given in µg/m^3^) for two US Embassy sites, from August 2016–December 2017, in Addis Ababa.

**Figure 4 ijerph-17-06998-f004:**
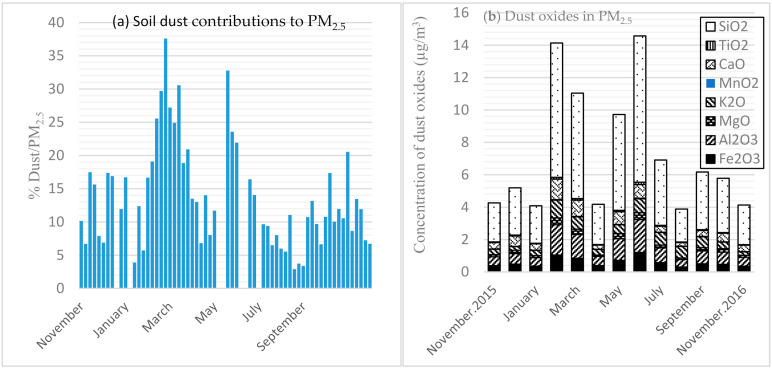
Dust oxides of fine particulate matter: (**a**) 1-in-6 days concentration of dust contribution to PM_2.5_ (Left) and (**b**) Monthly variation of dust oxides (Right) in Addis Ababa. N.B. openings in between daily dust oxide bars in Figure 4a were missing data: 24-December 2015, 16-May, 9-June, 15-June, and 3-July 2016 due to various reasons (including power outages), not days with zero values.

**Figure 5 ijerph-17-06998-f005:**
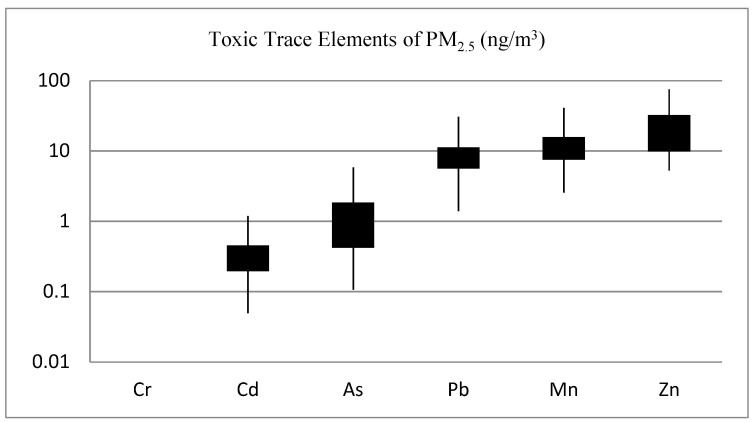
Concentration of toxic trace elements of PM_2.5_ in central Addis Ababa, Ethiopia.

**Table 1 ijerph-17-06998-t001:** Monthly average concentration of PM_2.5_ and bulk composition in Addis Ababa (µg/m^3^), Nov. 2015–Nov. 2016. (n is the number of observations per month).

Month (n)	Mass	EC ^a^	OM ^b^	SO_4_^2−^	NO_3_^−^	NH_4_^+^	OI ^c^	Dust	UM ^d^
Nov_15 (5)	39.82	8.36	22.83	2.59	0.32	0.64	0.38	4.26	0.44
Dec (4)	49.1	8.88	23.75	3.49	0.54	0.79	0.93	5.19	5.53
Jan (5)	36.89	8	20.12	2.8	0.6	0.46	0.26	4.08	0.57
Feb (5)	57.26	11.77	26.75	3.4	0.96	0.79	0.39	14.13	0
Mar (5)	45.1	9.63	21.66	2.25	0.19	0.7	0.31	11.04	0
April (5)	38.15	7.98	19.23	1.34	0.34	0.58	0.23	4.18	4.27
May (4)	44.53	11.61	15.55	1.71	0.39	0.46	1.22	9.72	3.87
June (3)	85.1	24.16	33.73	3.8	0.18	1.1	0.54	14.57	7.02
July (4)	79.91	21.75	36.23	2.89	0.26	1.25	0.64	6.91	9.98
Aug (5)	75.01	20.47	26.6	2.76	0.15	1.11	0.55	3.88	19.49
Sep (5)	77.17	22.83	31.28	5.25	0.27	2.68	0.98	6.17	7.71
Oct (6)	43.26	12.65	14.21	4.12	0.07	1.55	0.67	5.78	4.21
Nov_16 (5)	45.39	14.55	16.69	3.38	0.05	0.69	1.04	4.13	4.86

^a^ Elemental carbon. ^b^ Organic matter. ^c^ Other Ions, includes ws Na^+^, ws Cl^−^, ws K^+^, ws Ca^2+^. ^d^ Undetermined matter.

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
