# Peer review of "Chemical Characterization and Seasonality of Ambient Particles (PM2.5) in the City Centre of Addis Ababa"

_ijerph, 2020, doi:10.3390/ijerph17196998_

Round 1

Reviewer 1 Report

This paper describes an interesting set of measurements on the chemical composition of PM2.5 in Addis Ababa. The material is worth publishing. However, this article was not ready for submission. It contains many mistakes that should have been corrected through a careful proofreading before submission. I have listed a few issues below. I will be happy to review the contents of a revised manuscript.

1) Format the figures in a consistent manner. Use the same font and font size in all captions and legends. Do not use a colored background, e.g., Figure 5. Put the figures in the same order as they are discussed in the text, e.g., Figure 3 is after Figure 4. Confirm that the text refers to the correct figure, e.g., references to Figure 3 and Figure 4 are reversed. Use unit labels on all axes. Remove extraneous colored lines around figures, e.g., yellow lines in Figure 4b. Identify all symbols in each figure, e.g., circles in Figure 4b. The figure with WIOC, WSOC and EC has no figure number or caption. Figure 3 of the scatter plot is not referred to in the text. Remove notes to yourself from the figure captions, e.g., “(color could be used here)” in two captions.

2) Please group information on the same topic in the same paragraph. As one example, in the paragraph on lines 208-214, the first sentence is about sulfate and belongs with the preceding paragraph about sulfate.

3) Please check your numbers for consistency. How can OM be 40-63% (line 198) when the percentages in Figure 2b are ~25% to ~ 45%?

4) Please include the UM in Table 1.

5) Table S1 is referred to in the text but there is no Supplementary Information.

6) Check the formatting of numbers in the text. What does xxx.xx.x mean? +/- signs are missing in many places.

Author Response

Dear Reviewer#1

Thank you for the critical and educative comments you provided on this round.

We have addressed the comments and edits point by point responses as attached.

Thank you again for your time.

Worku Tefera

Reviewer 2 Report

The paper is interesting, which is a good and meaningful work. It can be piblished after minor revisions.

1. the topic shouold be more shorter and better.

2.the conclusion can be enhanced.

3.Some fresh work could be added to the references.

Author Response

Dear Reviewer#2

Thank you for the critical and educative comments you provided on this round for our manuscript on ambient particulate matter characterization.

We have addressed the comments and edits point by point responses as attached.

Thank you again for your time.

Worku Tefera

Reviewer 3 Report

Comments to Authors:

In this study titled "Characterization, Bulk Composition and Seasonality of Ambient Particulate Matter (PM2.5) in the city Center of Addis Ababa", authors attempted to characterize the annual average, bulk composition, and seasonal pattern of PM2.5 at central Addis Ababa based on observations collected during November, 2015 – November, 2016. The subject of this study is of interest to air quality researchers. Moreover, in the selected study area, there haven’t been many studies conducted in this region to characterize the composition and seasonal patterns of PM2.5. Based on this study, over 90% of the observed days, measured PM2.5 mass concentration exceeded WHO recommended standard.

In my opinion, this study needs additional work to discuss and explain the observed concentration, bulk composition, and seasonal characteristics of PM2.5. Also, I recommend the inclusion of meteorology data in this study (wind speed, wind direction, Temperature, relative humidity, rainfall) to explain the observed PM2.5 patterns. I recommend this manuscript to consider for publication after a major revision to accommodate for the aforementioned concerns and to address the following specific comments.

General Comments:

Grammar and Typos:
Proof read the manuscript thoroughly to correct for multiple typos.
Authors please follow a consistent verb tense throughout the manuscript and avoid mixed use of tenses.

Specific comments:

  • In my opinion, the authors conducted continuous PM5 measurements for about a year and analyzed these observed patterns or concentration. However, even though the authors provided some general explanations on rainfall and wind direction, to this reviewer that the current study lacks discussions connecting surface meteorology, topography, etc. on the daily, seasonal, and annual concentrations and compositions of PM2.5.  
  • This reviewer recommends the authors to analyze the ratio of OC / EC and include their discussion based on the ratio (Hint: OC/EC > 2 indicates secondary organic carbon aerosols formation)

Abstract:
The authors briefly discussed the observed concentration and characteristics of PM2.5. Authors wrote “PM2.5 in central Addis Ababa is dominated by EC and OC, which adversely affect the residents’ health”. As an example, the above discussion lacks a brief summary or author’s interpretation of how or why it going to adversely affect the residents’ health. For these reasons, the information presented in this abstract are insufficient to draw the interest of the reader for further reading. I recommend the authors revise this abstract.

Introduction:
Update the introduction section to provide sufficient background research and also include additional relevant references.

Material and Methods:
Avoid uncommon usages of meteorology terms. For example, throughout this manuscript, avoid using terms such as “small rainy”, “monsoonal rainy” etc.
Page #2, Line 70: Update “The most prevalent” to “The prevalent”

Results:
Page #4, Line 161: The average PM2.5 concentration is written here as 53.8 µg/m3 but Table-1 lists the annual average PM2.5 concentration as 55.13 µg/m3. Confirm and correct this at other places too in this manuscript.

Page #4, Line 167: update “minor rain season” to “season with light to moderate rain” or something like that.

Page #7, Lines 198 - 199: The authors wrote “The particulate OM was the dominant component of PM2.5 with contributions of 40.0%–63.0% for most days of the year”. What does it say about the major source of PM2.5 at this study area? Even though the concentration varied, the major source of pollution didn’t vary much?

Figure 3a: Figure3a of Carbonaceous content mass concentrations, there is a spike in the mass concentration of ~ 160.0 µg/m3 around early October. What is the cause of this? Was it a dust event or due to inversion condition?

Discussion:
This section must include additional detailed discussion on what are the major findings from this study and how is it compared to relevant previous research.

Page #10, Line 278: Authors claim that they didn’t find any previous studies on ambient PM2.5 concentration in Ethiopia. Even though not many, there are some previous studies conducted and are accessible online.

  • Trends of ambient air pollution and the corresponding respiratory diseases in addis ababa by Mekonnen Maschal Tarekegn and Tigist Yohannes, 2018
  • Mass concentrations and elemental composition of urban atmospheric aerosols in Addis Ababa, Ethiopia by G. Gebre, Z. Feleke, and E. Sahle-Demissie, 2010

Page #10, Lines 284 - 285: Please rewrite this sentence. Do you mean the annual average PM2.5 concentration 53.8 µg/m3? Is this value correct because the Table- 1 says the annual average value is 55.13 µg/m3. Confirm and correct this at other places too in this manuscript.

Page #10, Line 285: correct the typo “indicting” to “indicating”

Page #10, Line 285: correct “direct primary emissions” to “primary emissions”

Page #11, Line 305: wherever applicable, correct “high rainfall” to “heavy rainfall”

Conclusions:
What is/are the unique findings from this study? How important is this study in the ongoing air quality research in Addis Ababa? Please explain in detail, what are the future implications of this study?

Figures:

Some figures are not arranged or numbered correctly.
This reviewer suggest moving figure 2b to supplementary material because figure-2b conveys the same information as fig 2a but in %.
Figure1a: Figure1a caption says it is the trend of PM2.5 concentration during August 2016 through December 2017 but the x-axis indicates that it is during Nov. 2015 – Nov. 2016. Please update or clarify.

Figure 1b: suggesting to update the figure “diurnal pattern of PM2.5 concentration”  
Figure 2: One can easily identify the trends or information showed in figure3 in Figure2.   
Figure 3: Figure 3 of Dust oxides is numbered incorrectly as figure4 in line#236
Figure 4: Scatter plot of OC vs EC is numbered incorrectly as Figure3
Figure 5. The title of the figure 5 says it is “Toxic trace elements of PM ...” and it should be “Toxic trace elements of PM2.5 ...”

References:

Use a consistent reference style and add “doi” information to all possible referenced articles. Also follow the reference style recommended by this journal

The references need to be updated to include recent studies relevant to the research topic as well as the geographic area of this study.

Author Response

Dear Reviewer#3

Thank you for the critical, detail, and educative comments you provided on this round for our manuscript on PM2.5 characterization.

We have addressed the comments and edits point by point responses as attached. We have undertaken an academic edition for the manuscript to improve the language and grammar.

Thank you again for your time.

Worku Tefera

Round 2

Reviewer 1 Report

This paper describes an interesting set of measurements on the chemical composition of PM2.5 in Addis Ababa. The material is worth publishing. However, this article is poorly written and confusing. These measurements are not put in any sensible context. There are still many typos and formatting errors. The following comments about content need to be addressed by the authors before publication.
1) line 30. Do you mean “adopting” or “adapting effective prevention strategies.” This statement is repeated in the conclusions. If adapting, what adaptations are you proposing?
2) The description of the study setting is confusing (lines 73-76). The wind direction at the sampling site is mentioned and then the wind direction at the Bole area. How are those related to each other geographically? Why do we care about the Bole area? And how are they related to the two other sites for which data is presented? Late, the sampling site is described as central city (line 89), but it is unclear how that relates geographically to the Central and School monitors described in lines 201-204. A map would help, perhaps in the supplementary information. Also, please consolidate the description of the sampling location into one paragraph.
3) lines 153-155. This statement on “calculated chemical constituents” is confusing. Most of those are measured. Only dust oxides and POM are calculated.
4) lines 159-160. Why did you choose a conversion factor of 1.4 for OC to POM? Of the three references cited, Turpin et al. says 1.6 for urban aerosol, Russell et al. says 1.2-1.6, and Sardar et al. is not a reference for POM/OC. Please explain your choice.
5) Section 2.6 Data Analysis, lines 163-168. Delete this entire section. Lines 164-6 repeat lines 148-9 and lines 165-6 repeat line 159. It does not matter what graphing software you used.
6) lines 174-175, Why present both the annual average and the mean of the monthly averages? Does it tell you something that these are slightly different? It really only tells you that you had different numbers of days in each monthly average. Unless you are going to interpret the difference, it just seems confusing to present both. If there is a lot of variation in the number of observations per month, it might be worth putting that in parenthesis in Table 1.
7) Table 1 and Figure 2 are the same data. They should be presented and discussed together in the text. I would suggest reorganizing this section to present and discuss Table 1, Figure 2 and Figure 3 monthly data first, then move to the discussion of seasonal trends and the comparison to the BAM data.
8) Lines 200-209 and Figure 1. I find this figure hard to interpret. Since your goal is to compare seasonal trends, trying to compare hourly (?) BAM numbers with every few days filter data is difficult. It might make more sense to calculate the seasonal averages and compare those. Why do you include a diurnal of the BAM data? You never refer to Figure 1b in the text. If you are trying to use the diurnal trend to explain sources, like traffic, then you need to say that in the text. Otherwise, remove Figure 1b. I also think the discussion of seasonal trends would be helped by overlaying the seasons on Figure 2, perhaps with vertical bars and season labels.
9) Please stack the components in the same order in Figure S1 as in Figure 2 and use the same categories. It is very difficult to compare these two figures they way you are presenting them. Also, in Figure S1, it looks like OM is 25 – 45% of the total, but the text on line 229-230 says 35-56%. How can that be?
10) Lines 233-239, the amount of ammonium relative to the amount of sulfate and nitrate suggests that the aerosol was very acidic. Does this make sense?
11) lines 259-260, I don’t understand the statement about “forms SOA). Do you mean that the ratio of OC/EC > 2 indicates SOA? I think that would depend on source since primary BBOA also can have very high OC/EC.
12) lines 258-270, This paragraph is really confusing because it is mixing together many different concepts. What is the overall point that you are trying to make? It seems like you have a specific point based on your data that many days are unhealthy based on total mass loading. Then there’s the point that a small fraction of the days (11.5%) may have SOA. How is that related? Then there’s the relatively small ratio of WSOC/OC of 0.22 ± 0.18. Is that high enough to suggest high ROS activity and therefore health effects? Then there’s a general statement about carbonaceous matter and respiratory health. How is that related to your data? Or maybe these general statements are supposed to introduce the discussion in lines 278-290 of the actual results? Please reorganize this section so that you talk about one topic at a time and try to have a linear progression from you data to what it means.
13) lines 285-290. Why exclude these days? Aren’t they part of the air quality burden? Don’t they happen every year? Or maybe report the monthly averages with and without these particular days. Do these days really change the monthly averages?
14) lines 299-309. This paragraph is very repetitive. Lines 305 and 308 say the same thing. Lines 303 and 309 say the same thing. There is no reference to Figure 4b in the text. Why is it there?
15) lines 313-317 and Figure 5. Why is Cr included in Figure 5 when there is no point associated with it? Please list the elements in the same order in the text as they are in the figure. Table S2 is described as listing crustal factors, but it actually shows crustal enrichment factors. You need to explain what those are and what your source is for standard crustal composition. How can these elements in the ng/m3 range contribute a significant proportion of PM2.5 dust? There are orders of magnitude more crustal oxide.
16) The first half of the discussion section is really confusing. First there is a comparison with Delhi and Jeddah. Why these two cities? Then there is a much later paragraph with comparisons to a bunch of other cities. Why these cities? It is also really hard to interpret a string of numbers in text. I suggest that the authors first make a table of the numbers with the appropriate information about location (urban/rural), sampling duration (annual average, short campaign in what season), year, sources, etc.. Then see what patterns emerge from the table and discuss your results in the context of those patterns. I think it would make most sense to compare your results to other African cities first, and then to other continents.
17) Lines 347-352. I don’t understand the paragraph about the Tarekegn and Gulilat review. What is a desk review? What was the trend in particulate matter? How does this relate to your data?
18) Lines 353-361. You mention a satellite estimate. Is this good agreement with your data or not? Can you get seasonal estimates from the satellite data and do they agree with your data? I don’t understand the organization of this paragraph. How does the satellite number lead to a discussion of the number of cars? Shouldn’t that come after your discussion of your composition measurements?
19) Lines 362-368. What is the point of this discussion? Is it related to your data in some way? What is SOC?
20) Line 416. Start a new paragraph about WSOC.
Formatting, typographical and confusing language issues:
1) Please check the units on all numbers and use one version. For example, mass loading is expressed as ugm3 (incorrect, line 23), g m-3 (incorrect, line 193), ug/m3 and ug m-3. Please use the format suggested by the journal.
2) Figure S2 (line 80) is referred to in the text before Figure S1 (line 230). Figures should be in the same order as they are mentioned in the text.
3) line 91 “on the ground at a height of 2 m” is contradictory. Please rephrase.
4) line 94 define HOBO, include manufacturer
5) line 130 “Thermal Evolution/Optical Transmittance” is a technique not the name of an instrument. Don’t capitalize and do include the name of the instrument.
6) line 139, define USC CHS
7) Figure 2. Put the legend in the same order as the bars, e.g., EC is at the bottom of the bar so it should be at the bottom of the legend. Make sure that the caption and the legend agree in terms of cateogories.
8) line 218. The mean annual levels are not in either Table 1 or Figure 2. It would make sense to include them in Table 1 and then correct this sentence.
9) line 252-3, Table 1 is not percentages. Refer to Figure S1. The October percentage of OM looks like ~25%, not 33%.
10) line 258, you have switched from using ± to SD. Please use consistent terminology.
11) line 290, you already introduced Figure 3 in line 279. Why mention it here?

Author Response

Dear Reviewer#1

Thank you very much for your time and extensive and critical comments that helped this version of the manuscript to have a better look and more readable.

The authors addressed the comments by editing the manuscript and providing a point by point responses for each comment within the given time period.

We believe that the second round comments gave another opportunity to improve the manuscript by editing the typos extensively and provide additional description of the result and discussion parts to make this version better. We have also made structural adjustments for the readers to have a smooth transition and flow as per the suggestions.

Looking forward to hearing from you positively.

Stay safe,

Worku Tefera
